# Development of Algorithm for Clinical Management of Sickle Cell Bone Disease: Evidence for a Role of Vertebral Fractures in Patient Follow-up

**DOI:** 10.3390/jcm9051601

**Published:** 2020-05-25

**Authors:** Lucia De Franceschi, Daniele Gabbiani, Andrea Giusti, Gianluca Forni, Filippo Stefanoni, Valeria Maria Pinto, Giulia Sartori, Manuela Balocco, Chiara Dal Zotto, Maria Teresa Valenti, Luca Dalle Carbonare

**Affiliations:** 1Section of Internal Medicine, Department of Medicine, University of Verona, 37134 Verona, Italy; lucia.defranceschi@univr.it (L.D.F.); daniele.gabbiani@yahoo.com (D.G.); filippo.stefanoni@gmail.com (F.S.); giulia.sartori@gmail.com (G.S.); chiara.dalzotto@gmail.com (C.D.Z.); mariateresa.valenti@univr.it (M.T.V.); 2Rheumatology Unit, Department of Locomotor System, La Colletta Hospital, 16011 Arenzano, Italy; andreagiusti6613@gmail.com; 3Centro della Microcitemia, Anemie Congenite, Galliera Hospital, 16128 Genova, Italy; gianluca.forni@galliera.it (G.F.); valeriamaria.pinto@galliera.it (V.M.P.); manuela.balocco@galliera.it (M.B.)

**Keywords:** Sickle cell disease, bone densitometry, management, osteoporosis, vertebral fractures

## Abstract

Sickle-cell disease (SCD) is a worldwide distributed hemoglobinopathy, characterized by hemolytic anemia associated with vaso-occlusive events. These result in acute and chronic multiorgan damage. Bone is early involved, leading to long-term disability, chronic pain and fractures. Here, we carried out a retrospective study to evaluate sickle bone disease (SBD) in a cohort of adults with SCD. We assessed bone density, metabolism and turnover. We also evaluated the presence of fractures and the correlation between SCD severity and skeletal manifestations. A total of 71 patients with SCD were analyzed. The mean age of population was 39 ± 10 years, 56% of which were females. We found osteoporosis in a range between 7% and 18% with a high incidence of vertebral fractures. LDH and AST were predictive for the severity of vertebral fractures, while bone density was not. Noteworthy, we identified -1.4 Standard Deviations *T*-score as the cutoff for detecting the presence of fractures in patients with SCD. Collectively our data allowed us to develop an algorithm for the management of SBD, which may be useful in daily clinical practice to early intersect and treat SBD.

## 1. Introduction

Sickle-cell disease (SCD) is a worldwide distributed hemoglobinopathy characterized by hemolytic anemia associated with vaso-occlusive crisis, resulting in both acute and chronic organ damage. SCD is caused by a single gene mutation, leading to the production of pathologic sickle hemoglobin (HbS). SCD is defined for subjects homozygous HbS (SS) or heterozygous for HbS with co-inherited HbC (SC) or β-thalassemic trait (Sβ) [1].

Recurrent vaso-occlusive crises are the main cause of hospitalization of SCD patients and involve target organs such as spleen, brain, lung and bone. These are characterized by sluggish circulation, pro-oxidant environment, high oxygen extraction or relative local hypoxia. All factors favor HbS polymerization, intravascular sickling and increased cell-cell interactions with cell adhesion into microcirculation [2] [3,4,5,6]. In SCD, bone involvement has a very high prevalence, leading to long-term disability, chronic pain and fractures [7]. Avascular necrosis is one of the most common bone complication, involving femoral head in approximately one-third of adult patients with SCD [8]. This negatively impact patient quality of live, contributing to the comorbidity of adult patients with SCD [9].

Although SBD severely affects patients with SCD, the mechanisms involved in SBD are only partially known [2]. In a humanized mouse model for SCD, we recently show that SBD is generated by an unbalance between osteoclastogenesis and osteoblastogenesis, with a relative reduction in osteoblast recruitment and activity [2] This promotes an impairment of bone homeostasis, which is ameliorated by treatment with bisphosphonates such as zoledronic acid [2].

In patients with SCD, recurrent infarcts of bone and bone-marrow (BM), compensatory medullary hyperplasia associated with bone marrow niche vascular abnormality contribute local to insufficient oxygen and nourishment supply [2,10]. This results in intra BM sickling worsening local hypoxia, which may negatively impact mesenchymal stem cell compartment as well as the related osteoblast recruitment [11,12]. The biocomplexity of SBD is further increased by the presence of vitamin D deficiency, a condition very frequent also in the whole population, children and adults [13] which increased resorptive state and to a subsequent reduction in bone mineral density (BMD) [14,15]. In addition, sickle cell related kidney disease may negatively contribute to bone homeostasis, accelerating clinical manifestations of SBD [14,16]. Up to now limited data on therapeutic treatment of SBD are available with the exception of reports on vitamin D supplementation [17,18]. Treatment with either bisphosphonates, such as zoledronic acid, or denosumab, a human monoclonal IgG2 antibody, as bone anti-resorptive agents was discussed within SCD scientific community [19]. However, no clinical data are available yet.

In 2015, the Italian Society of Thalassemia and Hemoglobinophaties (SITE, www.site-italia.org) convened a panel of multidisciplinary experts to develop clinical practice recommendations to help physicians in the management of SBD [20,21].

Here, we first carried out a retrospective study on SBD in a cohort of SCD patients referring to two separate comprehensive sickle cell centers.

Longitudinal assessment of bone density, metabolism and turnover with the evaluation of the presence of fractures and the correlation between disease severity and skeletal manifestations were performed.

We then tested the SITE algorithm for clinical management of SBD, in two years after the publication of SITE recommendations for SBD and the final aim was to point out the predisposing factors to bone damage, identifying the best diagnostic pathway allowing an early, prompt intervention on SBD.

## 2. Methods

### 2.1. Patients and Design of the Study

We enrolled 71 sickle cell patients referring Verona and Genova (Italy) comprehensive centers for hemoglobinopathies from 2009 and 2017. All participants were informed on the aims of the study and signed the informed consent document before the enrollment. The study was registered in the Clinical Trial Registry (Clinicaltrial.gov ID: NCT02972138). The study complied with the revised ethical guidelines of the Declaration of Helsinki.

Diagnosis of SCD was confirmed by HPLC by the presence of either SS, SC or Sβ^0^. Forty-two percent of SCD patients was on stable HU treatment, 48% was on erythroapheresis and 10% was untreated.

Patients taking drugs that could alter bone metabolism such as corticosteroids, antiepileptics and thyroid hormones were excluded from the study. In the same way, post-menopausal women and patients affected by metabolic diseases such as primary hyperparathyroidism, chronic renal failure and liver failure were excluded from the study.

Biochemical and radiological data were collected during follow-up of SBD in 2009, 2012 and 2017. This window time was chosen based on the possibility to collect significant skeletal alterations, particularly regarding bone mass in such patients with small bone densitometric alterations at baseline. The flow-chart of the study is reported in Figure 1.

### 2.2. Clinical and Therapeutic Assessment

For each patient enrolled in the study, we collected patient history, which was annually updated with the number of severe vaso-occlusive crisis or acute chest syndrome (ACS).

We also recorded therapy for SBD such as vitamin D supplementation (cholecalciferol 100,000 IU per month), zoledronic acid or other bisphosphonates.

### 2.3. Biochemical and Radiological Assessment

In order to determine bone metabolism, serum calcium, phosphorus, alkaline phosphatase (ALP), PTH, 25-hydroxy vitamin D, urinary calcium excretion, C-terminal telopeptide of type I collagen (CTX, as a marker of bone resorption) and N-terminal pro-peptide of type I procollagen (P1NP, as a marker of osteoblastic activity) were measured. Bone density measurement and its progression over time were assessed with repeated measurements using dual-energy x-ray absorptiometry Hologic QDR Discovery Acclaim equipped with software for morphometric vertebral analysis (Hologic, Bedford, MA, USA). Lumbar spine (L1-L4) and proximal femur (neck and total hip) were the bone districts analyzed. The device was characterized by a coefficient of variation equal to 1.5% to the lumbar spine and 2.5% to the femur and a daily quality control was performed to assure accuracy and precision. World Health Organization (WHO) criteria for the diagnosis of osteoporosis were used (normal: T-score higher than −1 Standard Deviations (SD); osteopenia: T-score between −1 and −2.5 SD; osteoporosis: T-score lower than −2.5 SD, NHANES (National Health and Nutrition Examination Survey) normative database) [22]. In addition to morphometric vertebral analysis using bone densitometry, we collected thoracic and lumbar spine x-rays in lateral projection in order to identify vertebral fractures, in particular the typical “fish-shaped” vertebral bodies. For each vertebra we measured anterior, middle and posterior heights and classified fractures according to Genant grades: mild (20–25% height reduction), moderate (26–40% height reduction) and severe (> 40% height reduction) [22,23]. For Spine Deformity Index (SDI) calculation, a value of 0 was assigned to vertebrae without fractures, a value of 1 to mild fractures, a value of 2 to moderate fractures and a value of 3 to severe fractures; the sum of these values represents SDI index. SDI is a semi-quantitative index that integrates number and severity of fractures and was suggested as an indirectly surrogate marker of bone microarchitecture [24,25].

We also collected hematological data and biochemical parameters: Complete Blood Count, reticulocytes, ferritin, serum and urinary creatinine, glomerular filtration rate, transaminases (both aspartate and alanine aminotransferase), indirect bilirubin, lactate dehydrogenase (LDH).

## 3. Statistical Analysis

All statistical analyses were performed using Windows SPSS software, version 22.0 (SPSS, Inc., Chicago, IL, USA). Results obtained were expressed as mean ± standard deviation.

In order to analyze the differences between groups, Student’s *t*-test for paired data and multifactorial analysis of variance (ANOVA) were used.

We evaluated potential linear correlations between variables (Pearson’s coefficient) and potential predictive factors using multivariate linear regression tests (for continuous variables) and logistic regression (for categorical variables). ROC analysis was performed in order to evaluate the best cutoff of DXA expressed as *T*-score in terms of sensibility and specificity to predict the presence of vertebral fractures in this specific setting. Statistical significance was considered for a *p*-value of < 0.05.

## 4. Results

### 4.1. Population of the Study

A total of 71 patients with SCD were enrolled: 40 females and 31 males, with a mean age at the time of enrollment of 39 years. Among them, 22 (31%) were SS, 11 (15%) were SC and 38 (54%) were Sβ^0^, HbF was of 7.4%. Twenty-six patients (37%) were African, 35 (50%) Caucasian, 5 (7%) South American and 4 (6%) Central American. For one patient the ethnicity was unknown. Thirty-four patients (48%) were on erythroapheresis, 30 patients (42%) were on hydroxyurea (20 mg/Kg/d), and 7 patients (10%) were untreated. Concerning bone specific therapy 38 patients (53.5%) were taking vitamin D supplementation alone, whereas 9 (12.7%) were treated with bisphosphonates and vitamin D supplementation (Table 1).

The vaso-occlusive events recorded were defined as vaso-occlusive crisis (VOC) requiring hospitalization and acute chest syndrome (ACS). Forty-eight patients (68%) had presented at least one VOC in the observed period of time, 26 patients (37%) reported at least one ACS. All patients with ACS had experienced one VOC as well.

In SS patients, we found a higher number of VOC and ACS when compared to either SC or Sβ^0^ subjects (*p* < 0.05). In our population, we did not find any correlation between vitamin D levels and either VOC or ACS during the study.

### 4.2. Biochemical Evaluations

SCD patients showed a normocytic normochromic anemia with an average of Hb of 10.06 ± 0.10 g/dL (Table 2). RDW (18.34% ± 0.10%), reticulocytes (205.52 ± 66.14 × 10^9^/L), LDH (649.26 ± 26.67 *U*/L) and indirect bilirubin (1.72 ± 0.49 mg/dL) were increased, in agreement with chronic hemolysis, characterizing SCD. Hepatic and renal function were within the normal range.

We did not observe an effect of gender on hematologic and biochemical data. In SS patients, we found a significant reduction in Hct when compared to patients with either SC or Sβ^0^ genotype.

Noteworthy, platelet count (406 ± 145 × 10^9^/L *vs* 258 ± 94 × 10^9^/L; *p* < 0.005) and AST (42 ± 16 *U*/L *vs* 30 ± 10 *U*/L; *p* < 0.05) were higher in SCD patients with VOCs requiring hospitalization than in SCD individuals with VOCs managed at home.

Serum calcium, phosphate, ALP and urinary calcium excretion were normal. CTX levels were at the upper reference limits during the whole period of study. P1NP values were only available from 2016 and resulted within the normal range. No differences related to gender, genotype or therapies for SCD were found.

In agreement with previous reports [26], we observed hypovitaminosis D in 72% of SCD patients. Regarding 2009, we had data available for only 12 patients, 5 (41.6%) of which had values compatible with a deficiency condition (<10 ng/mL), while 7 (58.3%) presented levels within the insufficiency range (<30 ng/mL). In 2012, only one patient had adequate levels of vitamin D (> 30 ng/mL) and 26 patients (36.6%) had values compatible with vitamin D deficiency. In 2017, patients with vitamin D deficiency were 19 (26.7%) while adequate levels were found only in 9 subjects (12.7%).

Mean serum vitamin D level in the group of patients not taking any bone therapy was 14 ± 9 ng/mL, patients taking vitamin D supplementation showed the highest levels (22 ± 12 ng/mL), while the group taking bisphosphonates associated with vitamin D had the lowest levels (13 ± 12 ng/mL). Despite ongoing treatments, mean vitamin D levels of patients were insufficient.

Regarding vitamin D levels, no differences were found among genotypes, no correlations with liver function parameters and, evaluated by the logistic regression, no predictor factors were identified.

### 4.3. Bone Density

As for the evaluation of bone density in our population, out of 71 patients, 43 (58.9%) undergone a bone densitometry in 2017, 38 (52%) in 2012, while in 2009, 27 subjects (37%).

Along the study period, average bone mass values were normal (Table 3).

In 2009, bone densitometry showed lumbar osteoporosis in 2 patients (7.4%) with a mean BMD of 0.86 ± 0.02 g/cm^2^ and a *T*-score of −3.4 ± 0.6 SD; at femur level, only one subject showed osteoporosis (BMD 0.67 g/cm^2^ and *T*-score −3.2 SD).

Lumbar spine osteopenia was found in 7 patients (25.9%) with a BMD of 0.95 ± 0.05 g/cm^2^ and a *T*-score of −1.8 ± 0.2 SD; femur osteopenia was found in 4 patients (17.4%) (BMD 0.80 ± 0.03 g/cm^2^ and *T*-score −1.8 ± 0.4 SD).

In 2012 patients with lumbar osteoporosis were 7 (18.4%) with a BMD of 0.85 ± 0.04 g/cm^2^; at the femur level only one patient had osteoporosis (BMD 0.77 g/cm^2^ and *T*-score −2.5 SD).

Lumbar osteopenia was found in 7 subjects (18.4%) with a BMD of 0.97 ± 0.07 g/cm^2^ and a *T*-score of −1.4 ± 0.3 SD; femoral osteopenia was detected in 8 (22.2%) patients with a BMD of 0.83 ± 0.04 g/cm^2^ and a *T*-score of −1.5 ± 0.3 SD.

In 2017, we found lumbar osteoporosis in 7 patients (16.3%) with a mean BMD of 0.84 ± 0.07 g/cm^2^ and a *T*-score of −3.0 ± 0.1 SD. At femoral level, only one patient had osteoporosis with a BMD of 0.60 g/cm^2^ and a *T*-score of −3.8 SD. Lumbar osteopenia was found in 11 patients (25.6%) with a BMD of 1.02 ± 0.11 g/cm^2^ and a *T*-score—of 1.7 ± 0.3 SD; femur osteopenia was found in 9 subjects (22.5%) with a BMD of 0.84 ± 0.04 g/cm^2^ and a *T*-score of −1.5 ± 0.4 SD (Figure 2).

In 2012, we found 15 fractured patients (46.9%), the average number of fractures was 6 ± 2 with SDI score of 8 ± 4 (Figure 3). The patients were treated with a single infusion of zoledronic acid associated to cholecalciferol.

In 2017, the same 15 patients showed fractures, on average there were 7 ± 2 fractures per patient with SDI score of 10 ± 4. In order to investigate which factors may affect vertebral deformities, we compared patients with fractures with those with SDI score of zero. We found a significant difference between the two groups: AST (*p* < 0.005) and LDH (*p* < 0.05) were higher in fractured patients compared to unfractured subjects. Noteworthy, patients with vertebral fracture had higher femoral neck BMD and *T*-score (*p* < 0.05) compared to unfractured patients. No difference in lumbar between these two groups was observed.

To identify factors that may lead to an increase in fractures overtime, we compared SCD patients who had a stable SDI score with those with an increased SDI score from 2012 to 2017 (SCD with worsening SBD throughout the study. We found that MCH, reticulocytes and LDH were significantly higher in the group with worsened SDI score and increased vertebral fractures (*p* < 0.05). We also observed higher AST and ALT in SCD patients increased SDI (*p* < 0.05). In addition, SCD patients with increased SDI score over the time of observation had more VOCs compared to individuals with stable SDI score (*p* < 0.01). Noteworthy, we did not find any correlation between bone mineral density and vitamin D and worsening spine fractures.

We used logistic regression in order to determine which predictors of vertebral fractures in SCD patients studied. After correction for age, lumbar BMD and bone metabolism parameters, independent predictors of fractures were femoral neck BMD, *T*-Score and AST. Similarly, independent predictors of bone worsening were reticulocytes, LDH and AST.

We found no correlation between densitometric data and ethnicity, therapies performed or genotype, indicating that BMD is not a good predictor of bone impairment in SCD. To confirm this observation, we performed a ROC analysis using lumbar spine *T*-score as test variable (Figure 4). The AUC for this diagnostic tool in this setting was 0.67 confirming the low accuracy of this test in this setting (<0.7) and the best cutoff was −1.4 SD, showing 75% of sensitivity and 48% of specificity with a Youden’s index of 0.27. 

Finally, we observed no correlation between vitamin D levels and other metabolic or bone parameters, nor vitamin D was a predictor of skeletal disease. We then divided SCD patients according to bone treatments (vitamin D supplements, vitamin D supplements *plus* bisphosphonates or no therapy). Surprisingly, we found that 25(OH)D levels were higher in SCD patients taking only vitamin D supplementation compared to those under vitamin D supplementation *plus* bisphosphonate.

## 5. Discussion

SBD is chronic and invalidating complication of SCD. To date, limited data are available on clinical management of SBD. This is extremely important especially in perspective of possible novel antiresorptive such as denosumab or anabolic drugs such as antisclerostin antibodies developed for the treatment of fragility fractures [27]. In our cohort of SCD patients, CTX and P1NP were stably increased, supporting an active bone turnover in agreement with previous studies [28,29]. ALP was within normal range, suggesting that bone formation was not inhibited.

In patients with SCD, we confirmed the presence of vitamin D deficiency, despite the oral vitamin D supplementation [30]. Different factors have been invoked to explain the persistent vitamin D deficiency in SCD patients independently from vitamin D supplementation such as (i) the dark skin type present in 50% of our patients that may hinder the effect of solar radiation in vitamin D production [31]; (ii) reduced vitamin D intestinal absorption, most likely related to chronic cholestasis which may affect enterohepatic circulation of bile acids with reduced absorption of liposoluble compounds such as vitamin D [32]; (iii) increased body fat tissue levels in SCD patients could further induce vitamin D deficit by sequestration, thus determining its low bioavailability [12,33,34,35]. In our cohort, we showed that 72% of patients had vitamin D level <30 ng/mL and that hypovitaminosis D was confirmed even after supplementation. Thus, our results suggest that in SCD patients the dosage of vitamin D supplementation should be re-evaluated. In addition, we have to take into accounts the possible low compliance of SCD for vitamin D supplementation as supported by vitamin D plasma levels of SCD patients under bisphosphonates plus vitamin D supplementation similar to those of untreated SCD patients [36,37]. Interestingly, high prevalence of hypovitaminosis D may affect CTX levels, which were indeed increase in SCD patients leading to high bone turnover [38]. De Luna et al. [8] found lower CTX levels in SCD patients with high BMD, supporting the role of osteoclast activity [39].

A novel finding of our study is the low prevalence of osteoporosis based on bone mineral density in patients with SBD; in contrast, high prevalence of fractures and vertebral deformities in spine radiographs was found. Thus, the evaluation of spine radiographs, which allow the direct identification and characterization of skeletal alterations, represents a tool more powerful than BMD for the diagnosis and the follow-up of patients with SBD. Noteworthy, abnormal femoral neck *T*-score and BMD emerged as fracture predictors in subjects with SCD. Indeed, femoral neck, due to its functional anatomy and sluggish circulation, is a skeletal district prone to VOCs with development of avascular necrosis [40,41,42]. This aspect could explain the conflicting result of increased BMD as predictor of fracture. In our cohort, SCD patients with vertebral fractures show a more severe hemolytic phenotype defined by LDH level when compared to SCD subjects without vertebral fractures. In addition, in SCD patients we observed a significant correlation between abnormal AST and vertebral fracture, suggesting a possible role of sickle cell related hepatopathy in SBD. Previous studies in other models of hepatic osteodystrophy demonstrated the relationship between hepatic function and bone metabolism, vitamin D levels and bone density [43,44,45]. Inflammatory vasculopathy and chronic hemolysis play a key role in both acute and chronic sickle cell related organ complication [2]. Here, we found that SCD patients with bone disease worsening during the study have increase reticulocyte count, LDH. These are associated with higher prevalence of VOCs compared to SCD patient with stable SBD, supporting link between local ischemic-reperfusion damage, inflammatory vasculopathy and the severity of bone disease.

Take together our data allowed us to update the SITE algorithm for clinical management of SBD [21] including bone densitometry with the new cutoff suggested by ROC analysis in combination with spine X-ray (Figure 5).

Our study has some limitations since it is retrospective. Nevertheless, we firstly report data on a long-term follow-up on SBD and we develop an algorithm for clinical management of SBD, which represents a powerful tool to be tested in larger population of adult patients with SCD.

## 6. Conclusions

SCD is a complex hemoglobinopathy characterized by hemolytic anemia associated with chronic and acute multiorgan involvement. Bone is one of the affected organs and, due to increasing long-term survival of patients, represents an important target for prevention and therapy. The early assessment of bone involvement can prevent severe consequences of bone disease, mainly fractures. New findings emerged from the present study contribute to ameliorate the management of SCD patients, further increasing their survival and quality of life.

## Figures and Tables

**Figure 1 jcm-09-01601-f001:**
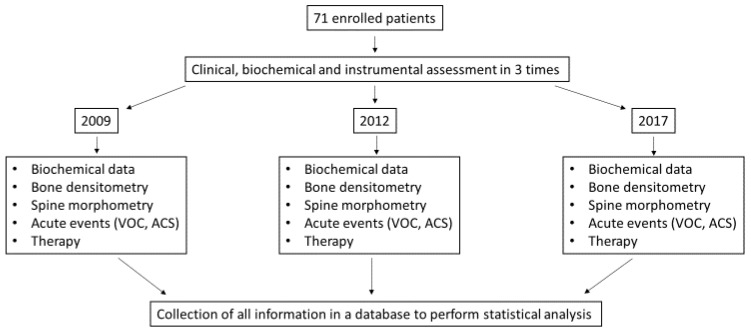
Flow-chart of the study. VOC: vaso-occlusive crisis; ACS: acute chest syndrome.

**Figure 2 jcm-09-01601-f002:**
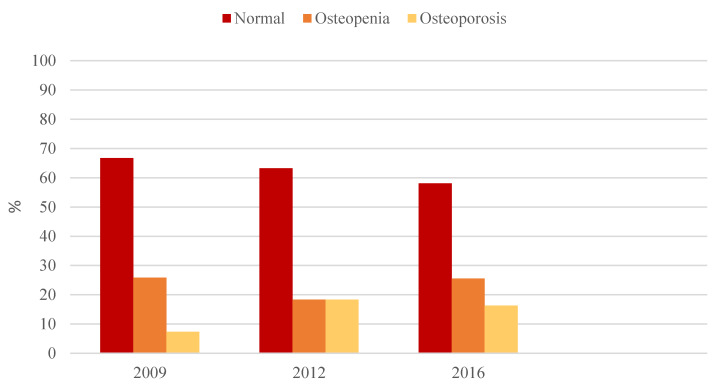
Lumbar bone mineral density grouped according to the World Health Organization (WHO) classification during the study period.

**Figure 3 jcm-09-01601-f003:**
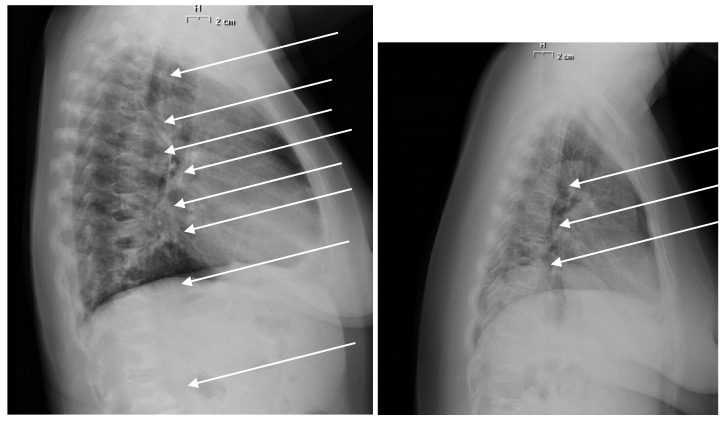
Thoracic x-ray of patients with vertebral fractures (arrows).

**Figure 4 jcm-09-01601-f004:**
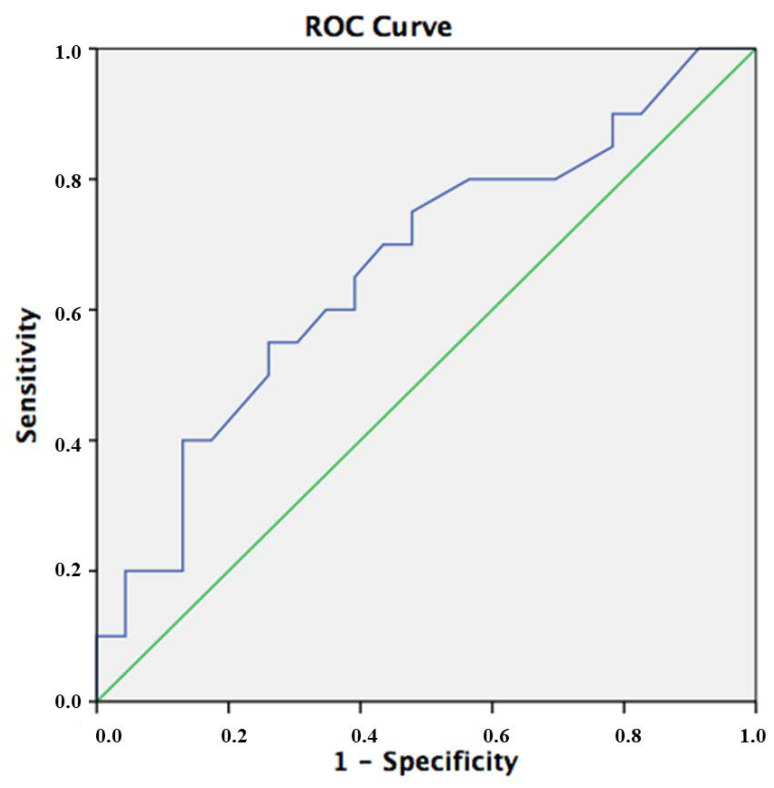
ROC (Receiver Operating Characteristic) curve analysis evaluating the sensitivity and specificity of lumbar bone mineral density (BMD) *vs* presence of fractures in SCD patients.

**Figure 5 jcm-09-01601-f005:**
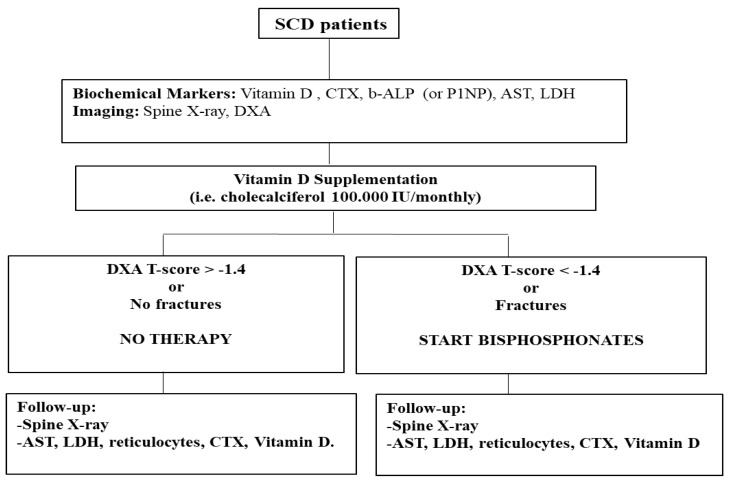
Algorithm for clinical management of sickle cell bone disease. CTX: carboxy-terminal collagen crosslinks; b-ALP: bone alkaline phosphatase; P1NP: procollagen type 1 N-terminal propeptide; AST: aspartate aminotransferase; LDH: lactate dehydrogenase; DXA: dual x-ray absorptiometry; SCD: sickle cell disease.

**Table 1 jcm-09-01601-t001:** Demographic data of SCD patients.

Male	31 (44%)
Female	40 (56%)
Age (yrs)	38.99 ± 10.26
SS	22 (31%)
SC	11 (15%)
Sβ^0^	38 (53%)
African	26 (37%)
Caucasian	35 (50%)
South American	5 (7%)
Central American	4 (6%)
Erytroapheresis	34 (48%)
Hydroxyurea	30 (42%)
Untreated pts for SCD	7 (10%)
Vitamin D supplementation	38 (53%)
Vitamin D + Bisphosphonates	9 (13%)
No bone therapy	24 (34%)

SS: homozygous HbS; SC: co-inherited HbC; Sβ^0^: β-thalassemic trait; SCD: Sickle Cell Disease.

**Table 2 jcm-09-01601-t002:** Hematologic and biochemical data of patients with SCD patients.

	2009	2012	2017
Erythrocytes (10^12^/L)	3.80 ± 0.70	3.88 ±0.57	3.75 ± 0.58
Hematocrit (%)	31.66 ± 4.06	32.56 ± 3.64	31.51 ± 3.38
Hemoglobin (g/dL)	10.57 ± 1.36	10.76 ± 1.13	10.52 ± 1.23
MCV (fL)	85.38 ± 9.72	86.81 ± 9.95	85.31 ± 9.24
MCH (pg)	28.44 ± 4.00	28.27 ± 3.34	28.35 ± 3.76
MCHC (g/dL)	40.20 ± 14.17	32.56 ± 1.05	31.92 ± 2.59
RDW (%)	–	18.43 ± 1.99	18.24 ± 2.30
Reticulocytes (10^9^/L)	225.68 ± 217.98	106.31 ± 137.95	284.57 ± 132.54
Platelets (10^9^/L)	399 ± 104.61	412.21 ± 133.67	375.99 ± 116.87
Leukocytes (10^9^/L)	9.67 ± 2.24	9.58 ± 2.97	9.93 ± 3.25
AST (U/L)	40.06 ± 16.28	36.36 ± 12.03	38.50 ± 15.15
Indirect bilirubin (mg/dL)	2.46 ± 1.35	1.35 ± 0.68	1.36 ± 0.78
LDH (U/L) (135–225)	634.25 ± 190.52	624.26 ± 131.95	689.27 ± 266.48
ALT (U/L) (6–50)	32.53 ± 16.10	31.34 ± 12.72	28.38 ± 12.93
Ferritin (mcg/L) (30–400)	–	1097.50 ± 899.32	1194.74±1097.75
Creatinine (mg/dL) (0.59–1.29)	0.71 ± 0.18	0.67 ± 0.16	0.80 ± 0.31
Calcium (mg/dL)(8.41–10.42)	9.23 ± 0.34	9.38 ± 0.32	9.12 ± 0.42
Phosphates (mg/dL) (2.63–4.49)	4.02 ± 0.5	3.52 ± 0.42	4.41 ± 1.80
Calcium/Creatinine (<0.57)	–	0.11 ± 0.07	0.35 ± 0.18
ALP (U/L) (50–130)	77.95 ± 22.74	76.00 ± 24.94	77.3 ± 27.46
CTX (ng/mL) (0100–0700)	0.44 ± 0.04	0.50 ± 0.21	0.47 ± 0.17
P1NP (mcg/L)(28–128)	–	–	64.81 ± 24.31
PTH (pg/mL) (1.6–6.9)	64.9 ± 11.72	33.67 ± 13.53	43.05 ± 22.10
Vitamin D (ng/mL) (>30)	14.05 ± 6.65	13.34 ± 6.33	17.65 ± 9.64

MCV: mean corpuscular volume; MCH: mean corpuscular hemoglobin; MCHC: mean cell hemoglobin concentration; RDW: red blood cell distribution width; LDH: Lactate dehydrogenase; AST: aspartate aminotransferase; ALT: alanine aminotransferase; ALP: alkaline phosphatase; CTX: carboxy-terminal collagen crosslinks; P1NP: procollagen type 1 N-terminal propeptide; PTH: parathyroid hormone.

**Table 3 jcm-09-01601-t003:** Densitometric data of SCD patients.

	2009	2012	2017
Spine BMD (g/cm^2^)	1.10 ± 0.15	1.12 ± 0.17	1.16 ± 0.21
Spine T score (SD)	−0.50 ± 1.38	−0.27 ± 1.50	−0.33 ± 1.68
Spine Z score (SD)	−0.27 ± 1.43	−0.04 ± 1.54	−0.07 ± 1.64
Femur total BMD (g/cm^2^)	1.01 ± 0.16	1.01 ± 0.15	1.01 ± 0.14
Femur total T score (SD)	−0.22 ± 1.10	−0.22 ± 0.99	−0.30 ± 0.98
Femur total Z score (SD)	−0.08 ± 1.15	−0.04 ± 1.01	−0.01 ± 0.98
Femur neck BMD (g/cm^2^)	0.89 ± 0.12	0.88 ± 0.11	0.87 ± 0.12
Femur neck T score (SD)	−0.76 ± 0.91	−0.70 ± 0.85	−1.03 ± 0.78
Femur neck Z score (SD)	−0.42 ± 0.88	−0.39 ± 0.84	−0.61 ± 0.80

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
