# Peer review of "Development of Algorithm for Clinical Management of Sickle Cell Bone Disease: Evidence for a Role of Vertebral Fractures in Patient Follow-up"

_jcm, 2020, doi:10.3390/jcm9051601_

Round 1

Reviewer 1 Report

The aim of the article by Di Franceschi et al is to highlight predisposing factors of chronic bone complications in patients with sickle cell disease.

However, the study raises several questions:

1/Methodologically, the authors specify that this is both a retrospective and prospective study since the same patients were evaluated in 2009, 2012 and 2017. Nevertheless, information is missing concerning the follow-up of patients between these periods of analysis, particularly regarding their treatment. For patients classified as “treated”, were they treated (whether with HU or Vitamin D for example) prior to inclusion? Were they treated for the all duration of the study? if not, what was the minimum duration of treatment deemed necessary to be considered as "treated"? It seems difficult to imagine that a sickle cell patient has never received vitamin D supplementation for 8 years, especially during the winter months...similarly, have some patients not switched from one category to another during the follow-up period? It is difficult to believe that any untreated patient remained untreated for the all duration of the study, if this is the case it should be stated whether patients changing treatment or initiating treatment were excluded from the study and the number of such patients should be stated with a new flow-chart.

2/ in the Verone patient group, it is very surprising to see that heterozygous AS patients were included in the study. Heterozygous AS is not a sickle cell disease and should not be confused with other forms of SCD. These "patients" have no reason to be included in the study and probably biases the results.

3/ Considering the low average HbS level in the cohort, it seems that the majority of thalassemic patients must have a thalassemic B+/S form. Once again, mixing these patients who can sometimes be almost assimilated to heterozygotes (if their Beta thalassemic trait is very moderate) with Beta° thalassemic/S or SS patients requires justification and detail.

4/ The authors rightly point out that chronic bone damage in patients with sickle cell disease is probably of multifactorial origin. However, several biological and clinical factors have not been collected. Clinically, how many patients developed osteonecrosis? Why have excluded patients with renal impairment knowing that it may be a risk factor? The BMI of the cohort that could have reflected the undernutrition and/or hypermetabolism of the patients is also missing, especially since BMI is known to have positive correlation with 1.25(OH)2D levels. Finally, with regard to iron overload, the mean ferritin level of the patients is surprisingly high (Table 4). Can you explain this iron overload? Other biological and radiological parameters could have been reported in these patients and then discussed. Are some of them being treated with iron chelators, which could impact their bone metabolism?

5/ several correlations (notably with lumbar bone density) are made with the number of erythrocytes. Usually the depth of anaemia is rather evaluated by the haemoglobin level, especially since in patients with a thalassaemic trait (which represents half of the cohort in this study), pseudo-polyglobulism can be observed. Why didn't you simply report hemoglobin levels? The red blood cell count is not the most relevant parameter for judging the depth of anemia in this context. In Figure 4, if you remove the AS subjects who likely have normal red blood cell counts and BML, does the correlation remain true?

6/ The authors report a "negative correlation between lumbar BMD and MCV". The impact of treatment with HU should definitely be discussed. This would imply that patients treated with HU, therefore with a higher MCV have a lower BMD? is this not counter-intuitive? 

7/ 15 fractured patients were found in 2012 and 15 patients showed fractures in 2017. Are these the same patients? If yes, what was their management between the 2 periods?

8/ In Table 3, the reticulocyte count is much lower in 2012. Please explain.

9/ The HbF rate of the patients could also have been interesting to analyze.

10/ In Table 4, the Z score is +0.08 in 2009. Is this a typo?

11/On the form finally, the manuscript is not very well constructed, with many results present in the discussion paragraph. The text also includes a few typos with expressions more Latin than Anglo-Saxon: for example the term eritro-exchange should be replaced by erythro-exchange, hydrossiurea (l.92, table 1) by hydroxyurea and drepanocitosis (Table 1 among others) by SCD, acute thoracic syndromes (l 173) by ACS, etc.

12/ Table 5 reports the main differences between unfractured and fractured patients. Are these differences significant? If so, please report the results of the statistical analysis in the table. The authors report a higher rate of AST in the fractured group and argue that "this finding can have a great clinical impact" and that "AST could therefore be a useful parameter for monitoring disease status". However, the values reported are almost all within normal ranges of AST for both groups.  It is difficult to extrapolate on a possible pathological predictive factor when the values are almost normal.

13/ "In addition to AST, also femoral neck T-score and BMD emerged as fractures predictors: SCD patients with vertebral fractures had significantly higher BMD and T-score at femoral level than unfractured". Be careful with this kind of conclusion. If we push the authors' reasoning, it would mean that a normal T-score is a risk factor for fracture, and therefore a low T-score would be Protective? Difficult to adhere to this kind of conclusion without more pathophysiological arguments.

14/ At the end of the discussion, the authors propose a new sub-group of patients, called "worsened patients". How are they defined? How are "stable" patients defined?  Table 6 does not add much insofar as it reports unexplained clinical categories and again does not include any statistical analysis.

Author Response

We are thankful to the referee for the comments on our manuscript. We’ll do our best to improve it according to the referee’s suggestions.

1/Methodologically, the authors specify that this is both a retrospective and prospective study since the same patients were evaluated in 2009, 2012 and 2017.

Nevertheless, information is missing concerning the follow-up of patients between these periods of analysis, particularly regarding their treatment.

As suggested we have rewritten part of method section. We hope that it is now more clear.

For patients classified as “treated”, were they treated (whether with HU or Vitamin D for example) prior to inclusion? Were they treated for the all duration of the study? if not, what was the minimum duration of treatment deemed necessary to be considered as "treated"?

Treated patients are referred to bone treatments, we specified specific SCD related treatment for HU or erythroapheresis. We have now clarified it in the text.

It seems difficult to imagine that a sickle cell patient has never received vitamin D supplementation for 8 years, especially during the winter months...similarly, have some patients not switched from one category to another during the follow-up period? It is difficult to believe that any untreated patient remained untreated for the all duration of the study, if this is the case it should be stated whether patients changing treatment or initiating treatment were excluded from the study and the number of such patients should be stated with a new flow-chart.

As suggested, we have excluded the untreated patients from the analysis.

2/ in the Verone patient group, it is very surprising to see that heterozygous AS patients were included in the study. Heterozygous AS is not a sickle cell disease and should not be confused with other forms of SCD. These "patients" have no reason to be included in the study and probably biases the results.

As suggested, we have removed the AS patients from the analysis

3/ Considering the low average HbS level in the cohort, it seems that the majority of thalassemic patients must have a thalassemic B+/S form. Once again, mixing these patients who can sometimes be almost assimilated to heterozygotes (if their Beta thalassemic trait is very moderate) with Beta° thalassemic/S or SS patients requires justification and detail.

The low average of HbS was affected by the data in SCD patients under erythroapheresis. We have removed this data from the analysis.

4/ The authors rightly point out that chronic bone damage in patients with sickle cell disease is probably of multifactorial origin. However, several biological and clinical factors have not been collected. Clinically, how many patients developed osteonecrosis? Why have excluded patients with renal impairment knowing that it may be a risk factor? The BMI of the cohort that could have reflected the undernutrition and/or hypermetabolism of the patients is also missing, especially since BMI is known to have positive correlation with 1.25(OH)2D levels.

As suggested, we have now added data on BMI and hip osteonecrosis. We have re-analyzed the data.

Finally, with regard to iron overload, the mean ferritin level of the patients is surprisingly high (Table 4). Can you explain this iron overload? ……Are some of them being treated with iron chelators, which could impact their bone metabolism?

In SCD, ferritin appears non-linear compared to liver iron loading as measured by MR (Adamkiewicz TV et al Blood 114: 4632, 2009) and it might be affected by chronic inflammation. In our patients, ferritin was increased compared to normal rage but stable throughout the study. In addition, we did not find iron accumulation in bone from humanized sickle cell mice when compared to bone from mouse model of b-thalassemia (Dalle Carbonare L et al Blood 126: 2320, 2015 see supplementary data)                                                                                                                          LIC is available for the SCD patients. LIC data might be added based if the reviewer believes it is important.

5/ several correlations (notably with lumbar bone density) are made with the number of erythrocytes. Usually the depth of anaemia is rather evaluated by the haemoglobin level, especially since in patients with a thalassaemic trait (which represents half of the cohort in this study), pseudo-polyglobulism can be observed. Why didn't you simply report hemoglobin levels? The red blood cell count is not the most relevant parameter for judging the depth of anemia in this context.

In Figure 4, if you remove the AS subjects who likely have normal red blood cell counts and BML, does the correlation remain true?

Thank you for this observation, we agree with the reviewer and we have removed correlations with the number of erythrocytes and figure 4.

6/ The authors report a "negative correlation between lumbar BMD and MCV". The impact of treatment with HU should definitely be discussed. This would imply that patients treated with HU, therefore with a higher MCV have a lower BMD? is this not counter-intuitive? 

We agree with the reviewer that the correlation between MCV and BMD might be affected by different factors in particular the presence of patients either on HU or erythroaphereis treatment. Thus, we removed the sentence.

7/ 15 fractured patients were found in 2012 and 15 patients showed fractures in 2017. Are these the same patients? If yes, what was their management between the 2 periods?

Yes, the patients were the same. They were treated with a single infusion of zoledronic acid associated to cholecalciferol.

8/ In Table 3, the reticulocyte count is much lower in 2012. Please explain.

The fluctuation of reticulocytes might be part hematologic phenotype in presence of chronic hemolysis. The difference between reticulocyte count at the different check-points did not reach the statistical significance.

9/ The HbF rate of the patients could also have been interesting to analyze.

As suggested, we have added the HbF data. No correlations were found with densitometric parameters.

10/ In Table 4, the Z score is +0.08 in 2009. Is this a typo?

 We thank the reviewer and we corrected the typo.

11/On the form finally, the manuscript is not very well constructed, with many results present in the discussion paragraph. The text also includes a few typos with expressions more Latin than Anglo-Saxon: for example the term eritro-exchange should be replaced by erythro-exchange, hydrossiurea (l.92, table 1) by hydroxyurea and drepanocitosis (Table 1 among others) by SCD, acute thoracic syndromes (l 173) by ACS, etc.

We thank the reviewer. We have reorganized the paper based on comments from both reviewer 1 and 2. We have also check typos and English throughout the text.

12/ Table 5 reports the main differences between unfractured and fractured patients. Are these differences significant? If so, please report the results of the statistical analysis in the table. The authors report a higher rate of AST in the fractured group and argue that "this finding can have a great clinical impact" and that "AST could therefore be a useful parameter for monitoring disease status". However, the values reported are almost all within normal ranges of AST for both groups.  It is difficult to extrapolate on a possible pathological predictive factor when the values are almost normal.

As suggested, we have rewritten the sentences in both result and discussion sections.

13/ "In addition to AST, also femoral neck T-score and BMD emerged as fractures predictors: SCD patients with vertebral fractures had significantly higher BMD and T-score at femoral level than unfractured". Be careful with this kind of conclusion. If we push the authors' reasoning, it would mean that a normal T-score is a risk factor for fracture, and therefore a low T-score would be Protective? Difficult to adhere to this kind of conclusion without more pathophysiological arguments.

We have better discussed the abnormal femoral neck BMD as predictor of fracture in SCD could be due to its functional anatomy and sluggish circulation, predisposing to VOCs with development of avascular necrosis. We hope to have clarified the point, now.

14/ At the end of the discussion, the authors propose a new sub-group of patients, called "worsened patients". How are they defined? How are "stable" patients defined?  Table 6 does not add much insofar as it reports unexplained clinical categories and again does not include any statistical analysis.

As suggested, we removed Table 6 and defined patients with worsened SBD.

Reviewer 2 Report

This is a descriptive study of changes to the bones in patients with all kinds of sickle cell disease over an 8-year period. Bone disease is an important SCD morbidity that, as the authors indicate, leads to significant pain and reduced quality of life. This is an understudied and important problem.

There are some English challenges that are surrmountable and that need to be addressed, for example, "hydrossiurea" should be re-spelled hydroxyurea and eritro-exchange is usually called red cell exchange transfusion (RCE). thalasso-drepanocytosis is usually called hemoglobin S-beta thalassemia zero or hemoglobin S-beta thalassemia plus. The use of prepositions ("of", "for") needs to be revisited. I suggest that the authors engage an editor to help with the language challenges (totally fixable).

Methods:

The authors need to clarify: Was this a prospective cohort study? Was it a convenience sample? How do they justify their sample size? How was 8-years of follow-up chosen? Why was a 3-year interval of follow-up chosen? Overall, the rational for the study design is absent and it's not clear what it contributes to the rigor of an observational study. Why differences did they hypothesize that they would see. The state aim of the study is to measure the impact of SCD on bones - suggest making this much narrower and more specific to address the specific and important measures used in this study.

Results:

Were the patients "selected" or enrolled based on meeting eligibility criteria?

Figure 2 appears to show the number of patients receiving no therapy, Vit D or Bisphosphonates + D. There is no "relationship" shown in this graph so it needs a new title. I'm not clear why this figure is needed as the numbers are clear and stated in the text.

The total study population should be described together. Perhaps a table could highlight differences between the two centers. The issue of number of African patients may be relevant as Vit D levels vary among genetically diverse populations.

Table 1: What is the difference between HbS heterozygous and HbSC? The categories African, Caucasian, South American, Central American are not parallel. African, South American and Central American are geographic locations. Caucasian isn't. This demographic information must be reconsidered; it cannot be presented like this.

Figure 3: In other studies does osteopenia and osteoporosis progress in measurable ways over 3 year intervals?

Discussion: Needs to be revised to highlight the most significant, novel findings. Discussion should focus on the pathobiological, clinical or management considerations raised by the data.

It is strange that patients were followed for 8 years and there was no assessment of Vit D adherence even though the authors stigmatize SCD patients by stating that adherence is a common problem. Indeed if, as the authors indicate, Vit D levels do not correlate with bone disease, is it rational for patients to take vitamin d or irrational to prescribe it?

There is a rather extensive literature on differences in Vit D levels based on skin color - skin melatonin and genetics -it is curiously missing from the discussion.

In the abstract, the authors conclude that AST and LDH are associated with bone density, but offer no discussion of how hemolysis and hemolytic rate might affect bones.

Overall: This paper needs to be significant revised to help the reader identify the primary aim, primary analysis and primary results. The rational for the approach needs to be provided and the discussion focused on the most important findings.

Author Response

We are thankful to the referee for the comments on our manuscript. We’ll do our best to improve it according to the referee’s suggestions.

  1. There are some English challenges that are surrmountable and that need to be addressed, for example, "hydrossiurea" should be re-spelled hydroxyurea and eritro-exchange is usually called red cell exchange transfusion (RCE). thalasso-drepanocytosis is usually called hemoglobin S-beta thalassemia zero or hemoglobin S-beta thalassemia plus. The use of prepositions ("of", "for") needs to be revisited. I suggest that the authors engage an editor to help with the language challenges (totally fixable).

As suggested we have largely revised the text for typos and English.

  1. Methods:

The authors need to clarify: Was this a prospective cohort study? Was it a convenience sample? How do they justify their sample size? How was 8-years of follow-up chosen? Why was a 3-year interval of follow-up chosen?

Overall, the rational for the study design is absent and it's not clear what it contributes to the rigor of an observational study……..The state aim of the study is to measure the impact of SCD on bones - suggest making this much narrower and more specific to address the specific and important measures used in this study.

As suggested we have rewritten method section and aims of the study. 3-year of interval follow-up was chosen based on the possibility to collect significant skeletal alterations, particularly regarding bone mass in such patients with small bone densitometric alterations at baseline.

  1. Results:
  • Were the patients "selected" or enrolled based on meeting eligibility criteria?

The patients were enrolled consecutively with the exception of patients with exclusion criteria.

  • Figure 2 appears to show the number of patients receiving no therapy, Vit D or Bisphosphonates + D. There is no "relationship" shown in this graph so it needs a new title. I'm not clear why this figure is needed as the numbers are clear and stated in the text.

As suggested, we have removed Figure 2.

  • The total study population should be described together. Perhaps a table could highlight differences between the two centers.

As suggested we have described the 2 population together.

  • The issue of number of African patients may be relevant as Vit D levels vary among genetically diverse populations.

Table 1: What is the difference between HbS heterozygous and HbSC? The categories African, Caucasian, South American, Central American are not parallel. African, South American and Central American are geographic locations. Caucasian isn't. This demographic information must be reconsidered; it cannot be presented like this.

As suggested we have refined data presentation.

Figure 3: In other studies does osteopenia and osteoporosis progress in measurable ways over 3 year intervals?

As stated in the discussion, bone densitometry is not a good predictor for the progression of osteoporosis and we suggest a X-ray evaluation of the spine.

Discussion:

  • Needs to be revised to highlight the most significant, novel findings. Discussion should focus on the pathobiological, clinical or management considerations raised by the data….. Overall: This paper needs to be significant revised to help the reader identify the primary aim, primary analysis and primary results. The rational for the approach needs to be provided and the discussion focused on the most important findings.

As suggested we extensively revised discussion section.

  • It is strange that patients were followed for 8 years and there was no assessment of Vit D adherence even though the authors stigmatize SCD patients by stating that adherence is a common problem. Indeed if, as the authors indicate, Vit D levels do not correlate with bone disease, is it rational for patients to take vitamin d or irrational to prescribe it?.....There is a rather extensive literature on differences in Vit D levels based on skin color - skin melatonin and genetics -it is curiously missing from the discussion.

As suggested, we extensively revised discussion section and added new sentences concerning vitamin D.

  • In the abstract, the authors conclude that AST and LDH are associated with bone density, but offer no discussion of how hemolysis and hemolytic rate might affect bones.

As suggested, we added few sentences in discussion section on the link between hemolysis, ischemic-reperfusion damage and SBD.

Round 2

Reviewer 1 Report

The revised manuscript is quite difficult to read given the significant changes and apparent modifications. The whole thing remains quite drafty, with apparent comments, unnumbered figures, duplicate graphs...However, the work seems to have been completely revised, which is a good thing.

 The message is now simpler and clearer.

Nevertheless, some editing work remains necessary before the manuscript is presentable.